# Learning to Screen for Fast Softmax Inference on Large Vocabulary Neural Networks

**Pei-Hung (Patrick) Chen**[1]*, **Si Si**[2], **Sanjiv Kumar**[2], **Yang Li**[2], **Cho-Jui Hsieh**[12]
[1]Department of Computer Science, University of California, Los Angeles
[2]Google Research
`patrickchen@g.ucla.edu`, `{sisidaisy,sanjivk,liyang}@google.com`,
`chohsieh@cs.ucla.edu`

## Abstract

Neural language models have been widely used in various NLP tasks, including machine translation, next word prediction and conversational agents. However, it is challenging to deploy these models on mobile devices due to their slow prediction speed, where the bottleneck is to compute top candidates in the softmax layer. In this paper, we introduce a novel softmax layer approximation algorithm by exploiting the clustering structure of context vectors. Our algorithm uses a light-weight screening model to predict a much smaller set of candidate words based on the given context, and then conducts an exact softmax only within that subset. Training such a procedure end-to-end is challenging as traditional clustering methods are discrete and non-differentiable, and thus unable to be used with back-propagation in the training process. Using the Gumbel softmax, we are able to train the screening model end-to-end on the training set to exploit data distribution. The algorithm achieves an order of magnitude faster inference than the original softmax layer for predicting top-$k$ words in various tasks such as beam search in machine translation or next words prediction. For example, for machine translation task on German to English dataset with around 25K vocabulary, we can achieve 20.4 times speed up with 98.9% precision@1 and 99.3% precision@5 with the original softmax layer prediction, while state-of-the-art (Zhang et al., 2018) only achieves 6.7x speedup with 98.7% precision@1 and 98.1% precision@5 for the same task.

## 1 Introduction

Neural networks have been widely used in many natural language processing (NLP) tasks, including neural machine translation (Sutskever et al., 2014), text summarization (Rush et al., 2015) and dialogue systems (Li et al., 2016). In these applications, a neural network (e.g. LSTM) summarizes current state by a context vector, and a softmax layer is used to predict the next output word based on this context vector. The softmax layer first computes the "logit" of each word in the vocabulary, defined by the inner product of context vector and weight vector, and then a softmax function is used to transform logits into probabilities. For most applications, only top-$k$ candidates are needed, for example in neural machine translation where $k$ corresponds to the search beam size. In this procedure, the computational complexity of softmax layer is linear in the vocabulary size, which can easily go beyond 10K. Therefore, the softmax layer has become the computational bottleneck in many NLP applications at inference time.

Our goal is to speed up the prediction time of softmax layer. In fact, computing top-$k$ predictions in softmax layer is equivalent to the classical Maximum Inner Product Search (MIPS) problem—given a query, finding $k$ vectors in a database that have the largest inner product values with the query. In neural language model prediction, context vectors are equivalent to queries, and weight vectors are equivalent to the database. MIPS is an important operation in the prediction phase of many machine learning models, and many algorithms have been developed (Bachrach et al., 2014; Shrivastava & Li, 2014; Neyshabur & Srebro, 2015; Yu et al., 2017; Guo et al., 2016). Surprisingly, when we apply

---

*This work is conducted during Pei-Hung Chen's internship in Google Research.

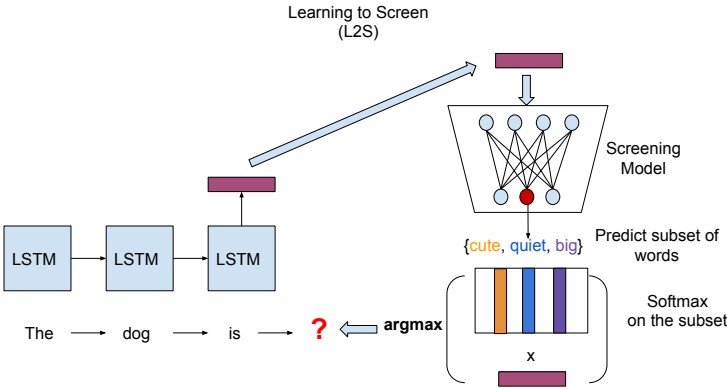

Figure 1: Illustration of the proposed algorithm.

recent MIPS algorithms to LSTM language model prediction, there's not much speedup if we need to achieve $> 98\%$ precision (see experimental section for more details). This motivates our work to develop a new algorithm for fast neural language model prediction.

In natural language, some combinations of words appear very frequently, and when some specific combination appears, it is almost-sure that the prediction should only be within a small subset of vocabulary. This observation leads to the following question: *Can we learn a faster "screening" model that identifies a smaller subset of potential predictions based on a query vector?* In order to achieve this goal, we need to design a learning algorithm to exploit the distribution of context vectors (queries). This is quite unique compared with previous MIPS algorithms, where most of them only exploit the structure of database (e.g., KD-tree, PCA-tree, or small world graph) instead of utilizing the query distribution.

We propose a novel algorithm (L2S: learning to screen) to exploit the distribution of both context embeddings (queries) and word embeddings (database) to speed up the inference time in softmax layer. To narrow down the search space, we first develop a light-weight screening model to predict the subset of words that are more likely to belong to top-$k$ candidates, and then conduct an exact softmax only within the subset. The algorithm can be illustrated in Figure 1. Our contribution is four folds:

- We propose a screening model to exploit the clustering structure of context features. All the previous neural language models only consider partitioning the embedding matrix to exploit the clustering structure of the word embedding to achieve prediction speedup.
- To make prediction for a context embedding, after obtaining cluster assignment from screening model, L2S only needs to evaluate a small set of vocabulary in that cluster. Therefore, L2S can significantly reduce the inference time complexity from $O(Ld)$ to $O((r + \bar{L})d)$ with $\bar{L} \ll L$ and $r \ll L$ where $d$ is the context vector' dimension; $L$ is the vocabulary size, $r$ is the number of clusters, and $\bar{L}$ is the average word/candidate size inside clusters.
- We propose to form a joint optimization problem to learn both screening model for clustering as well as the candidate label set inside each cluster simultaneously. Using the Gumbel trick (Jang et al., 2017), we are able to train the screening network end-to-end on the training data.
- We show in our experiment that L2S can quickly identify the top-$k$ prediction words in the vocabulary an order of magnitude faster than original softmax layer inference for machine translation and next words prediction tasks.

## 2 RELATED WORK

We summarize previous works on speeding up the softmax layer computation.

**Algorithms for speeding up softmax in the training phase.** Many approaches have been proposed for speeding up softmax training. Jean et al. (2014); Mnih & Teh (2012) proposed importance sampling techniques to select only a small subset as "hard negative samples" to conduct the updates. The hierarchical softmax-based methods (Morin & Bengio, 2005; Grave et al., 2017) use the tree

structure for decomposition of the conditional probabilities, constructed based on external word semantic hierarchy or by word frequency. Most hierarchical softmax methods cannot be used to speed up inference time since they only provide a faster way to compute probability for a target word, but not for choosing top-$k$ predictions as they still need to compute the logits for all the words for inference. One exception is the recent work by Grave et al. (2017), which constructs the tree structure by putting frequent words in the first layer—so in the prediction phase, if top-$k$ words are found in the first layer, they do not need to go down the tree. We provide comparison with this approach in our experiments.

**Algorithms for Maximum Inner Product Search (MIPS).** Given a query vector and a database with $n$ candidate vectors, MIPS aims to identify a subset of vectors in the database that have top-$k$ inner product values with the query. Top-$k$ softmax can be naturally approximated by conducting MIPS. Here we summarize existing MIPS algorithms:

- Hashing: (Shrivastava & Li, 2014; Neyshabur & Srebro, 2015) proposed to reduce MIPS to nearest neighbor search (NNS) and then solve NNS by Locality Sensitive Hashing (LSH) (Indyk & Motwani, 1998).
- Database partitioning: PCA tree (Sproull, 1991) partitions the space according to the directions of principal components and shows better performance in practice. Bachrach et al. (2014) shows tree-based approaches can be used for solving MIPS but the performance is poor for high dimensional data.
- Graph-based algorithm: Malkov et al. (2014); Malkov & Yashunin (2016a) recently developed an NNS algorithm based on small world graph. The main idea is to form a graph with candidate vectors as nodes and edges are formed between nearby candidate vectors. The query stage can then done by navigating in this graph. Zhang et al. (2018) applies the MIPS-to-NNS reduction and shows graph-based approach performs well on neural language model prediction.
- Direct solvers for MIPS: Some algorithms are proposed to directly tackle MIPS problem instead of transforming to NNS. Guo et al. (2016); Wu et al. (2017) use quantization-based approach to approximate candidate set. Another Greedy MIPS algorithm is recently proposed in (Yu et al., 2017), showing significant improvement over LSH and tree-based approaches.

**Algorithms for speeding up softmax in inference time.** MIPS algorithms can be used to speed up the prediction phase of softmax layer, since we can view context vectors as query vectors and weight vectors as database. In the experiments, we also include the comparisons with hashing-based approach (LSH) (Indyk & Motwani, 1998), partition-based approach (PCA-tree (Sproull, 1991)) and Greedy approach (Yu et al., 2017). The results show that they perform worse than graph-based approach (Zhang et al., 2018) and are not efficient if we want to keep a high precision.

For NLP tasks, there are two previous attempts to speed up softmax layer prediction time. (Shim et al., 2017) proposed to approximate the weight matrix in the softmax layer with singular value decomposition, find a smaller candidate set based on the approximate logits, and then do a fine-grained search within the subset. (Zhang et al., 2018) transformed MIPS to NNS and applied graph-based NNS algorithm to speed up softmax. In the experiments, we show our algorithm is faster and more accurate than all these previous algorithms. Although they also have a screening component to select an important subset, our algorithm is able to learn the screening component using training data in an end-to-end manner to achieve better performance.

## 3 ALGORITHM

Softmax layer is the main bottleneck when making prediction in neural language models. We assume $L$ is the number of output tokens, $W \in \mathbb{R}^{d \times L}$ is the weight matrix of the softmax layer, and $\boldsymbol{b} \in \mathbb{R}^L$ is the bias vector. For a given context vector $\boldsymbol{h} \in \mathbb{R}^d$ (such as output of LSTM), softmax layer first computes the logits

$$x_s = \boldsymbol{w}_s^T \boldsymbol{h} + b_s \quad \text{for} \quad s = 1, \cdots, L \tag{1}$$

where $\boldsymbol{w}_s$ is the $s$-th column of $W$ and $b_s$ is the $s$-th entry of $\boldsymbol{b}$, and then transform logits into probabilities $p_s = \frac{e^{x_s}}{\sum_{l=1}^{L} e^{x_l}}$ for $s = 1, \cdots, L$. Finally it outputs the top-$k$ candidate set by sorting the probabilities $[p_1, \cdots, p_L]$, and uses this information to perform beam search in translation or predict next word in language model.

To speedup the computation of top-$k$ candidates, all the previous algorithms try to exploit the structure of $\{\boldsymbol{w}_s\}_{s=1}^{L}$ vectors, such as low-rank, tree partitioning or small world graphs (Zhang et al., 2018; Shim et al., 2017; Grave et al., 2017). However, in NLP applications, there exists strong structure of context vectors $\{\boldsymbol{h}\}$ that has not been exploited in previous work. In natural language, some combinations of words appear very frequently, and when some specific combinations appear, the next word should only be within a small subset of vocabulary. Intuitively, if two context vectors $\boldsymbol{h}_i$ and $\boldsymbol{h}_j$ are similar, meaning similar context, then their candidate label sets $C(\boldsymbol{h}_i)$ and $C(\boldsymbol{h}_j)$ can be shared. In other words, suppose we already know the candidate set of $C(\boldsymbol{h}_i)$ for $\boldsymbol{h}_i$, to find the candidate set for $\boldsymbol{h}_j$, instead of computing the logits for all $L$ tokens in the vocabulary, we can narrow down the candidate sets to be $C(\boldsymbol{h}_i)$, and only compute the logits in $C(\boldsymbol{h}_i)$ to find top-$k$ prediction for $\boldsymbol{h}_j$.

**The Prediction Process.** Suppose the context vectors are partitioned into $r$ disjoint clusters and similar ones are grouped in the same partition/cluster, if a vector $\boldsymbol{h}$ falls into one of the cluster, we will narrow down to that cluster's label sets and only compute the logits of that label set. This screening model is parameterized by clustering weights $\boldsymbol{v}_1, \ldots, \boldsymbol{v}_r \in \mathbb{R}^d$ and label candidate set for each cluster $\boldsymbol{c}_1, \ldots, \boldsymbol{c}_r \in \{0, 1\}^L$. To predict a hidden state $\boldsymbol{h}$, our algorithm first computes the cluster indicator

$$z(\boldsymbol{h}) = \arg\max_t \boldsymbol{v}_t^T \boldsymbol{h}, \tag{2}$$

and then narrows down the search space to $C(\boldsymbol{h}) := \{s \mid \boldsymbol{c}_{z(\boldsymbol{h}),s} = 1\}$. The exact softmax is then computed within the subset $C(\boldsymbol{h})$ to find the top-$k$ predictions (used in language model) or compute probabilities used for beam search in neural machine translation. As we can see the prediction time includes two steps. The first step has $r$ inner product operations to find the cluster which takes $O(rd)$ time. The second step computes softmax over a subset, which takes $O(\bar{L}d)$ time where $\bar{L}$ ($\bar{L} \ll L$) is the average number of labels in the subsets. Overall the prediction time for a context embedding $h$ is $O((r + \bar{L})d)$, which is much smaller than the $O(Ld)$ complexity using the vanilla softmax layer. Figure 1 illustrates the overall prediction process.

However, how to learn the clustering parameter $\{\boldsymbol{v}_t\}_{t=1}^{r}$ and the candidate sets $\{\boldsymbol{c}_t\}_{t=1}^{r}$? We found that running spherical kmeans on all the context vectors in the training set can lead to reasonable results (as shown in the appendix), but can we learn even parameters to minimize the prediction error? In the following, we propose an *end-to-end procedure to learn both context clusters and candidate subsets simultaneously* to maximize the performance.

**Learning the clustering.** Traditional clustering algorithms such as kmeans on Euclidean space or cosine similarity have two drawbacks. First, they are discrete and non-differentiable, and thus hard to use with back-propagation in the end-to-end training process. Second, they only consider clustering on $\{\boldsymbol{h}_i\}_{i=1}^{N}$, without taking the predicted label space into account. In this paper, we consider learning the partition through Gumbel-softmax trick. We will briefly summarize the technique and direct the reader to (Jang et al., 2017) for further details on these techniques. In Table 5 in the appendix, we compare our proposed method to traditional spherical-kmeans to show that it can further improve the performance.

First, we turn the deterministic clustering in Eq(2) into a stochastic process: the probability that $\boldsymbol{h}$ belongs to cluster $t$ is modeled as

$$P(t|\boldsymbol{h}) = \frac{\exp(\boldsymbol{v}_t^T \boldsymbol{h})}{\sum_{j=1}^{r} \exp(\boldsymbol{v}_j^T \boldsymbol{h})}, \ \forall t, \quad \text{and} \quad z(\boldsymbol{h}) = \arg\max_t P(t|\boldsymbol{h}). \tag{3}$$

However, since argmax is a discrete operation, we cannot combine this operation with final objective function to find out better clustering weight vectors. To overcome this, we can re-parameterize Eq(3) using Gumbel trick. Gumbel trick provides an efficient way to draw samples $\boldsymbol{z}$ from the categorical distribution calculated in Eq(3):

$$m(\boldsymbol{h}) = \text{one\_hot}(\arg\max_j [g_j + \log P(j|\boldsymbol{h})]), \tag{4}$$

where each $g_j$ is an i.i.d sample drawn from Gumbel$(0, 1)$. We then use the Gumbel softmax with temperature $= 1$ as a continuous, differentiable approximation to argmax, and generate $r$-dimensional sample vectors $\boldsymbol{p} = [p_1, \cdots, p_r]$ which is approximately one-hot $m(\boldsymbol{h})$ with

$$p_t = \frac{\exp(\log(P(t|\boldsymbol{h})) + g_t)}{\sum_{j=1}^{r} \exp(\log(P(j|\boldsymbol{h})) + g_j)}, \forall t \in \{1, \ldots, r\}. \tag{5}$$

Using the Straight-Through (ST) technique proposed in (Jang et al., 2017), we denote $\bar{\boldsymbol{p}} = \boldsymbol{p} +$ (one_hot($\arg\max_j \boldsymbol{p}_j$) $- \boldsymbol{p}$) as the one-hot representation of $\boldsymbol{p}$ and assume back-propagation only goes through the first term. This enables end-to-end training with the loss function defined in the following section. We also use $\bar{p}(h)$ to denote the one-hot entry of $\bar{\boldsymbol{p}}$ (i.e., the position of the "one" entry of $\bar{\boldsymbol{p}}$).

**Learning the candidate set for each cluster.** For a context vector $\boldsymbol{h}_i$, after getting into the partition $t$, we will narrow down the search space of labels to a smaller subset. Let $\boldsymbol{c}_t$ be the label vector for $t$-th cluster, we define the following loss to penalize the mis-match between correct predictions and the candidate set:

$$\ell(\boldsymbol{h}_i, \boldsymbol{y}_i) = \sum_{s:\boldsymbol{y}_{is}=1} (1 - \boldsymbol{c}_{ts})^2 + \lambda \sum_{s:\boldsymbol{y}_{is}=0} (\boldsymbol{c}_{ts})^2, \tag{6}$$

where $\boldsymbol{y}_i \in \{0,1\}^L$ is the 'ground truth' label vector for $\boldsymbol{h}_i$ that is computed from the exact softmax. We set $\boldsymbol{y}$ to be the label vector from full softmax because our goal is to approximate full softmax prediction results while having faster inference (same setting with (Shim et al., 2017; Zhang et al., 2018)). The loss is designed based on the following intuition: when we narrow down the candidate set, there are two types of loss: 1) When a candidate $s$ ($\boldsymbol{y}_{is} = 1$) is a correct prediction but not in the candidate set ($\boldsymbol{c}_{ts} = 0$), then our algorithm will miss this label. 2) When a candidate $j$ ($\boldsymbol{y}_{is} = 0$) is not a correct prediction but it's in the candidate set ($\boldsymbol{c}_{ts} = 1$), then our algorithm will waste the computation of one vector product. Intuitively, 1) is much worse than 2), so we put a much smaller weight $\lambda \in (0, 1)$ on the second term.

The choice of true candidate set in $\boldsymbol{y}$ can be set according to the application. Throughout this paper, we set $\boldsymbol{y}$ to be the correct top-5 prediction (i.e., positions of 5-largest $x_s$ in Eq(1). $\boldsymbol{y}_{is} = 1$ means $s$ is within the correct top-5 prediction of $\boldsymbol{h}_i$, while $\boldsymbol{y}_{is} = 0$ means it's outside the top-5 prediction.

**Final objective function:** We propose to learn the partition function (parameterized by $\{\boldsymbol{v}_t\}_{t=1}^r$) and the candidate sets ($\{\boldsymbol{c}_t\}_{t=1}^r$) simultaneously. The joint objective function will be:

$$\underset{\substack{\boldsymbol{v}_1,\cdots,\boldsymbol{v}_r \\ \boldsymbol{c}_1,\cdots,\boldsymbol{c}_r}}{\text{minimize}} \sum_{i=1}^N \Big( \sum_{s:\boldsymbol{y}_{is}=1} (1 - \boldsymbol{c}_{\bar{p}(\boldsymbol{h}_i),s})^2 + \lambda \sum_{s:\boldsymbol{y}_{is}=0} (\boldsymbol{c}_{\bar{p}(\boldsymbol{h}_i),s})^2 \Big) \tag{7}$$
$$\text{s.t. } \boldsymbol{c}_t \in \{0,1\}^L \ \forall t = 1,\ldots,r$$
$$\bar{L} \leq B \ \forall i = 1,\ldots,N$$

where $N$ is the number of samples, $\bar{L}$ is the average label size defined as $\bar{L} = (\sum_{i=1}^N \sum_{s=1}^L \boldsymbol{c}_{\bar{p}(\boldsymbol{h}_i),s})/N$, $\bar{p}(\boldsymbol{h}_i)$ is the index for where $\bar{\boldsymbol{p}}_t = 1$ for $t = 1,\cdots,r$; $B$ is the desired average label/candidate size across different clusters which could be thought as prediction time budget. Since $\bar{L}$ is related to the computation time of proposed method, by enforcing $\bar{L} \leq B$ we can make sure label sets won't grow too large and desired speed-up rate could be achieved. Note that $\bar{p}(\boldsymbol{h}_i)$ is for clustering assignment, and thus a function of clustering parameters $\boldsymbol{v}_1,\cdots,\boldsymbol{v}_r$ as shown in Eq(3).

**Optimization.** To solve the optimization problem in Eq (7), we apply alternating minimization. First, when fixing the clustering (parameters $\{\boldsymbol{v}_t\}$) to update the candidate sets (parameters $\{\boldsymbol{c}_t\}$), the problem is identical to the classic "Knapsack" problem—each $\boldsymbol{c}_{t,s}$ is an item, with weight proportional to number of samples belonging to this cluster, and value defined by the loss function of Eq(7), and the goal is to maximize the value within weight capacity $B$. There is no polynomial time solution with respect to $r$, so we apply a greedy approach to solve it. We sort items by the value-capacity ratio and add them one-by-one until reaching the upper capacity $B$.

When fixing $\{\boldsymbol{c}_t\}$ and learning $\{\boldsymbol{v}_t\}$, we convert the cluster size constraint to objective function by Lagrange-Multiplier:

$$\underset{\boldsymbol{v}_1,\cdots,\boldsymbol{v}_r}{\text{minimize}} \sum_{i=1}^N \Big( \sum_{j:y_{is}=1} (1 - \boldsymbol{c}_{\bar{p}(\boldsymbol{h}_i)s})^2 + \lambda \sum_{j:y_{is}=0} (\boldsymbol{c}_{\bar{p}(\boldsymbol{h}_i)s})^2 \Big) + \gamma \max(0, \bar{L} - B) \tag{8}$$
$$\text{s.t. } \boldsymbol{c}_t \in \{0,1\}^L \ \forall t = 1,\ldots,r,$$

and simply use SGD since back-propagation is available after applying Gumbel trick. To deal with $\bar{L}$ in the mini-batch setting, we replace it by the moving-average, updated at each iteration when we go through a batch of samples. The overall learning algorithm is given in Algorithm 1.

---

**Algorithm 1:** Training Process for Learning to Screen (L2S)

---

**Input**: Context vectors $\{h_i\}_{i=1}^{N}$ (e.g., from LSTM); trained network's softmax layer's weight $W$ and basis vector $b$.

**Output**: Clustering parameters $v_t$ and candidate label set $c_t$ for each cluster for $t = 1, \cdots, r$.

1 **Hyperparameter:** Number of clusters $r$; prediction time budget $B$; regularization terms $\lambda$ and $\gamma$; number of iterations $T$.

2 Compute ground true label vector $\{y_i\}_{i=1}^{N}$ with only top-$k$ non-zeros entries. The top-$k$ labels for each context vector $h_i$ are generated by computing and then sorting the values in $x_i = W^T h_i + b$;

3 Initialize cluster weights $\{v_t\}_{t=1}^{r}$ using spherical kmeans over $\{h_i\}_{i=1}^{N}$ ;

4 Initialize the label set for each cluster $\{c_t\}_{t=1}^{r}$ to be zeros;

5 **for** $j = 1, \cdots, T$ **do**

6      Fixing $\{c_t\}_{t=1}^{r}$ and learning the clustering parameters $\{v_t\}_{t=1}^{r}$ in Eq(8) by SGD with Gumbel trick;

7      Fixing $\{v_t\}_{t=1}^{r}$ and learning the labels set $c_t$ $t = 1, \cdots, r$ by solving the "Knapsack" problem using Greedy approach;

8 **return** $c_t, v_t$ for all $t = 1, \cdots, r$.

---

Table 1: The vocabulary sizes and hidden dimension used in each experiment.

| Models | vocabulary size | dimension |
|---|---|---|
| PTB-Small | 10000 | 200 |
| PTB-Large | 10000 | 1500 |
| WikiText-103 | 80000 | 400 |
| NMT: DE-EN | 24725 | 500 |
| NMT: EN-VE | 22749 | 200 |

## 4 EXPERIMENTS

We evaluate our method on two tasks: Language Modeling (LM) and Neural Machine Translation (NMT). For LM, we use the Penn Treebank Bank (PTB) dataset (Marcus et al., 1993). For NMT, we use the IWSLT 2014 German-to-English translation task (Cettolo et al., 2014) and IWSLT 2015 English-Vietnamese data (Luong & Manning, 2015). All the models use a 2-layer LSTM neural network structure. For IWSLT-14 DE-EN task, we use the PyTorch checkpoint provided by Open-NMT (Klein et al., 2017). For IESLT-15 EN-VE task, we set the dimension of hidden size to be 200, and the rest follows the default training hyperparameters of OpenNMT. For PTB, we train a 2-layer LSTM-based language model on PTB from scratch with two setups: PTB-Small and PTB-Large. The LSTM hidden state sizes are 200 for PTB-Small and 1500 for PTB-Large, so are their embedding sizes. Vocabulary sizes and the hidden dimension used in our experiments are summarized in Table 1.

We verified that all these models achieved benchmark performance on the corresponding datasets as reported in the literature. We then apply our method to accelerate the inference of these benchmark models.

### 4.1 COMPETING ALGORITHMS

We include the following algorithms in our comparisons:

- L2S (Our proposed algorithm): the proposed learning-to-screen method. Number of clusters and average label size across clusters will be the main hyperparameters affecting computational time. We could control the tradeoff of time and accuracy by fixing the number of clusters and varying the size constraint $B$. For all the experiments we set parameters $\lambda = 0.0003$ and $\gamma = 10$. We will show later that L2S is robust to different numbers of clusters.
- FGD (Zhang et al., 2018): transform the softmax inference problem into nearest neighbor search (NNS) and solve it by a graph-based NNS algorithm.

Table 2: Comparison of softmax prediction results on three datasets. Speedup is based on the original softmax time. For example, 10x means the method's prediction time is 10 times faster than original softmax layer prediction time. Computation of full softmax per step is 4.32 ms for PTB-Large, 0.32 ms for PTB-Small and 4.83 ms for NMT: DE-EN.

|  | PTB-Small | | | PTB-Large | | | NMT: DE-EN | | |
|  | Speedup | P@1 | P@5 | Speedup | P@1 | P@5 | Speedup | P@1 | P@5 |
|---|---|---|---|---|---|---|---|---|---|
| L2S (Our Method) | **10.6x** | **0.998** | **0.990** | **45.3x** | **0.996** | **0.982** | **20.4x** | **0.989** | **0.993** |
| FGD | 1.3x | 0.980 | 0.989 | 6.9x | 0.975 | 0.979 | 6.7x | 0.987 | 0.981 |
| SVD-softmax | 0.8x | 0.987 | 0.99 | 2.3x | 0.988 | 0.981 | 3.4x | 0.98 | 0.985 |
| Adaptive-softmax | 1.9x | 0.972 | 0.981 | 4.2x | 0.974 | 0.937 | 3.2x | 0.982 | 0.984 |
| Greedy-MIPS | 0.5x | 0.998 | 0.972 | 1.8x | 0.945 | 0.903 | 2.6x | 0.911 | 0.887 |
| PCA-MIPS | 0.14x | 0.322 | 0.341 | 0.5x | 0.361 | 0.326 | 1.3x | 0.379 | 0.320 |
| LSH-MIPS | 1.3x | 0.165 | 0.33 | 2.2x | 0.353 | 0.31 | 1.6x | 0.131 | 0.137 |

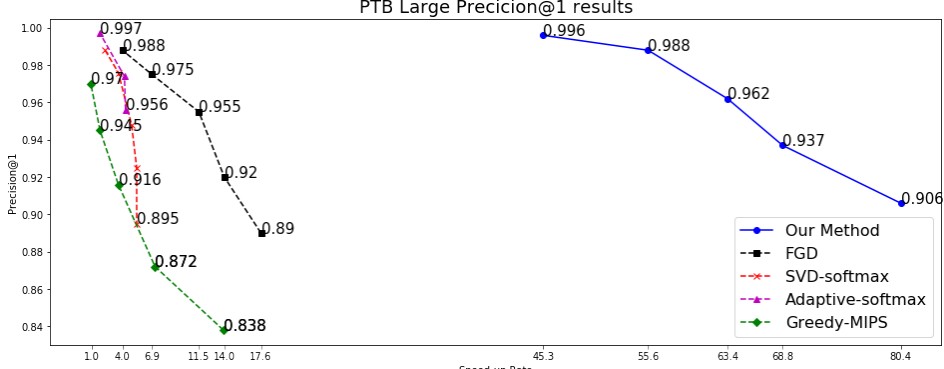

Figure 2: Precision@1 versus speed-up rate of PTB Large Setup.

- SVD-softmax (Shim et al., 2017): a low-rank approximation approach for fast softmax computation. We vary the rank of SVD to control the tradeoff between prediction speed and accuracy.
- Adaptive-softmax (Grave et al., 2017): a variant of hierarchical softmax that was mainly developed for fast training on GPUs. However, this algorithm can also be used to speedup prediction time (as discussed in Section 2), so we include it in our comparison. The tradeoff is controlled by varying the number of frequent words in the top level in the algorithm.
- Greedy-MIPS (Yu et al., 2017): the greedy algorithm for solving MIPS problem. The tradeoff is controlled by varying the budget parameter in the algorithm.
- PCA-MIPS (Bachrach et al., 2014): transform MIPS into Nearest Neighbor Search (NNS) and then solve NNS by PCA-tree. The tradeoff is controlled by varying the tree depth.
- LSH-MIPS (Neyshabur & Srebro, 2015): transform MIPS into NNS and then solve NNS by Locality Sensitive Hashing (LSH). The tradeoff is controlled by varying number of hash functions.

We implement L2S, SVD-softmax and Adaptive-softmax in numpy. For FGD, we use the C++ library implemented in (Malkov & Yashunin, 2016b; Boytsov & Naidan, 2013) for the core NNS operations. The last three algorithms (Greedy-MIPS, PCA-MIPS and LSH-MIPS) have not been used to speed up softmax prediction in the literature and they do not perform well in these NLP tasks, but we still include them in the experiments for completeness. We use the C++ code by (Yu et al., 2017) to run experiments for these three MIPS algorithms.

Since our focus is to speedup the softmax layer which is known to be the bottleneck of NLP tasks with large vocabulary, we only report the prediction time results for the softmax layer in all the experiments. To compare under the same amount of hardware resource, all the experiments were conducted on an Intel Xeon E5-2620 CPU using a single thread.

## 4.2 PERFORMANCE COMPARISONS

To measure the quality of top-$k$ approximate softmax, we compute Precision@$k$ (P@$k$) defined by $|A_k \cap S_k|/k$, where $A_k$ is the top-$k$ candidates computed by the approximate algorithm and $S_k$ is

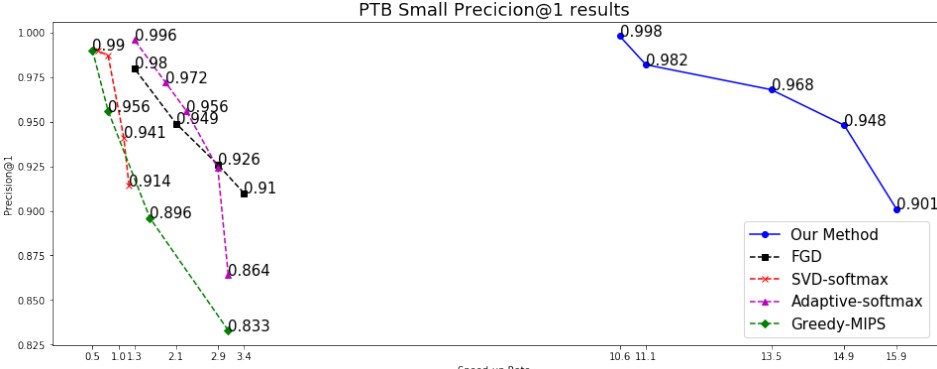

Figure 3: Precision@1 versus speed-up rate of PTB Small Setup.

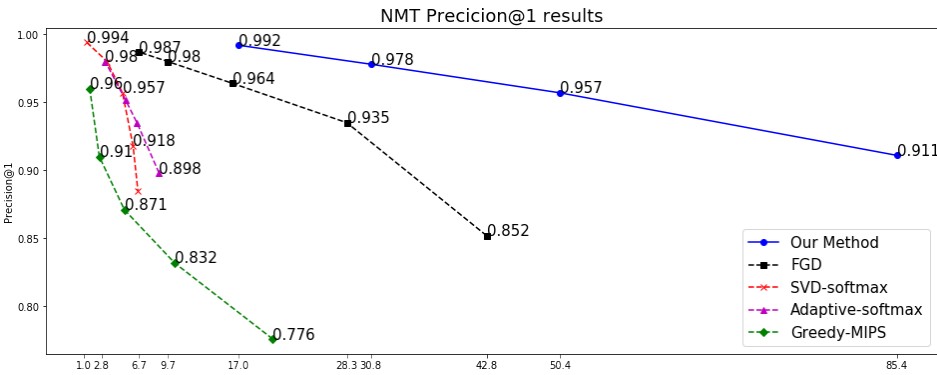

Figure 4: Precision@1 versus speed-up rate of NMT:DE-EN Setup.

the top-$k$ candidates computed by exact softmax. We present the results for $k = 1, 5$. This measures the accuracy of next-word-prediction in LM and NMT. To measure the speed of each algorithm, we report the speedup defined by the ratio of wall clock time of the exact softmax to find top-$k$ words divided by the wall clock time of the approximate algorithm.

For each algorithm, we show the prediction accuracy vs speedup over the exact softmax in Figure 2, 3, 4, 5, 6, 7 (The last three are in the appendix). We do not show the results for PCA-MIPS and LSH-MIPS in the figures as their curves run outside the range of the figures. Some represented results are reported in Table 2. These results indicate that the proposed algorithm significantly outperforms all the previous algorithms for predicting top-$k$ words/tokens on both language model (next word prediction) and neural machine translation.

Next, we measure the BLEU score of the NMT tasks when incorporating the proposed algorithm with beam search. We consider the common settings with beam size = 1 or 5, and report the wall clock time of each algorithm excluding the LSTM part. We only calculate log-softmax values on reduced search space and leave probability of other vocabularies not in the reduced search space to be 0. From the precision comparison, since FGD shows better performance than other completing methods in Table 2, we only compare our method with state-of-the-art algorithm FGD in Table 3 in terms of BLEU score. Our method can achieve more than 13 times speed up with only 0.14 loss in BLEU score in DE-EN task with beam size 5. Similarly, our method can achieve 20 times speed up in EN-VE task with only 0.08 loss in BLEU score. In comparison, FGD can only achieve less than 3-6 times speed up over exact softmax to achieve a similar BLEU score. We also compare our algorithm with other methods using perplexity as a metric in PTB-Small and PTB-Large as shown in Table 6 in the appendix. We observe more than 5 times speedup over using full softmax without losing much perplexity (less than 5% difference). More details can be found in the appendix.

In addition, we also show some qualitative results of our proposed method on DE-EN translation task in Table 8 to demonstrate that our algorithm can provide similar translation results but with faster inference time.

Table 3: Comparison of BLEU score results vs prediction time on DE-EN and EN-VE task. Speedup is based on the original softmax time.

| Model | Metric | Original | FGD | Our method |
|---|---|---|---|---|
| NMT: DE-EN | Speedup Rate | 1x | 2.7x | 14.0x |
| Beam=1 | BLEU | 29.50 | 29.43 | 29.46 |
| NMT: DE-EN | Speedup Rate | 1x | 2.9x | 13.4x |
| Beam=5 | BLEU | 30.33 | 30.13 | 30.19 |
| NMT: EN-VE | Speedup Rate | 1x | 6.4x | 12.4x |
| Beam=1 | BLEU | 24.58 | 24.28 | 24.38 |
| NMT: EN-VE | Speedup Rate | 1x | 4.6x | 20x |
| Beam=5 | BLEU | 25.35 | 25.26 | 25.27 |

Table 4: L2S with different number of clusters.

| Number of Clusters | 50 | 100 | 200 | 250 |
|---|---|---|---|---|
| Time in ms | 0.12 | 0.17 | 0.14 | 0.12 |
| P@1 | 0.997 | 0.998 | 0.998 | 0.994 |
| P@5 | 0.988 | 0.99 | 0.99 | 0.98 |

### 4.3 SELECTION OF THE NUMBER OF CLUSTERS

Finally, we show the performance of our method with different number of clusters in Table 4. When varying number of clusters, we also vary the time budget $B$ so that the prediction time including finding the correct cluster and computing the softmax in the candidate set are similar. The results indicate that our method is quite robust to number of clusters. Therefore, in practice we suggest to just choose the number of clusters to be 100 or 200 and tune the "time budget" in our loss function to get the desired speed-accuracy tradeoff.

## 5 CONCLUSION

In this paper, we proposed a new algorithm for fast softmax inference on large vocabulary neural language models. The main idea is to use a light-weight screening model to predict a smaller subset of candidates, and then conduct exact search within that subset. By forming a joint optimization problem, we are able to learn the screening network end-to-end using the Gumbel trick. In the experiment, we show that the proposed algorithm achieves much better inference speedup than state-of-the-art algorithms for language model and machine translation tasks.

## 6 ACKNOWLEDGEMENT

We are grateful to Ciprian Chelba for the fruitful comments, corrections and inspiration. CJH acknowledges the support of NSF via IIS-1719097, Intel faculty award, Google Cloud and Nvidia.

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

# 7 APPENDIX

## 7.1 PRECISION@5 RESULTS

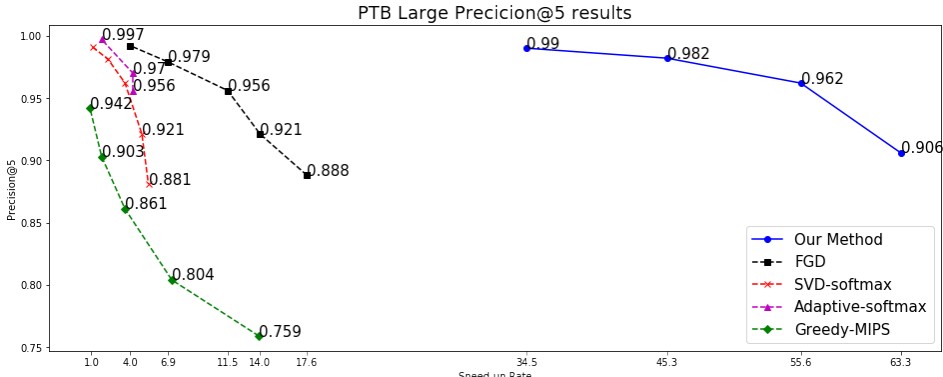

Figure 5: Precision@5 versus speed-up rate of PTB Large Setup.

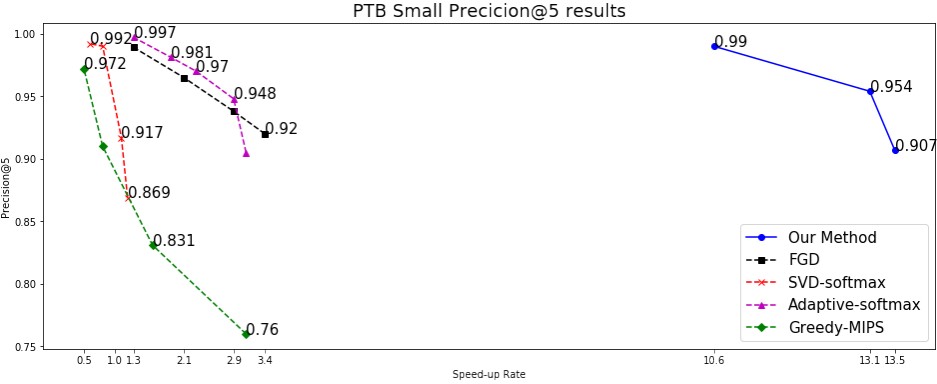

Figure 6: Precision@5 versus speed-up rate of PTB Small Setup.

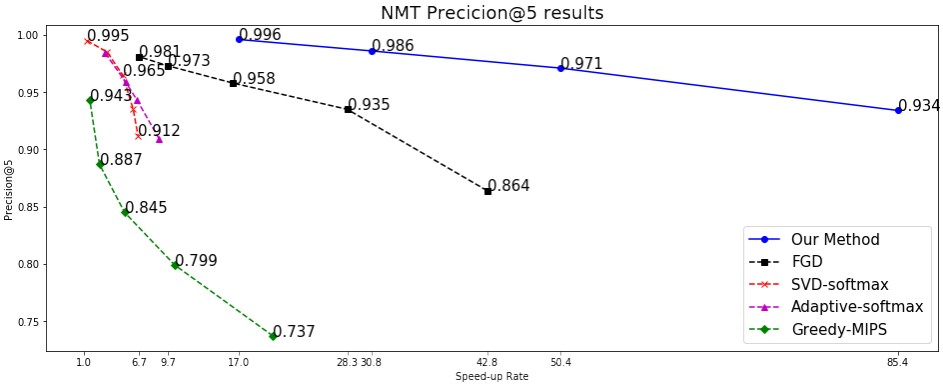

Figure 7: Precision@5 versus speed-up rate of NMT:DE-EN Setup.

## 7.2 COMPARISON TO SPHERICAL-KMEANS INITIALIZATION

Since we firstly initialize parameters in our method by Shperical-KMEANS, we also show in Table 5 that L2S can further improve over the baseline clustering methods. Notice that even the basic

Table 5: Comparison of L2S to spherical-KMEANS clustering.

| | PTB-Small | | | PTB-Large | | | NMT: DE-EN | | |
|---|---|---|---|---|---|---|---|---|---|
| | Speedup | P@1 | P@5 | Speedup | P@1 | P@5 | Speedup | P@1 | P@5 |
| Our Method | **10.6x** | **0.998** | **0.990** | **45.3x** | **0.999** | **0.82** | **20.4x** | **0.989** | **0.993** |
| Sphereical-kmeans | 4x | 0.988 | 0.992 | 6.9x | 0.992 | 0.971 | 13.8x | 0.991 | 0.993 |
| FGD | 1.3x | 0.980 | 0.989 | 6.9x | 0.975 | 0.979 | 6.7x | 0.987 | 0.981 |

Table 6: Comparison of Perplexity results vs prediction time on PTB dataset.

| Model | Metric | Original | SVD-softmax | Adaptive-softmax | FGD | Our method |
|---|---|---|---|---|---|---|
| PTB-Small | Speedup Rate | 1x | 0.84x | 1.69x | 0.95x | 5.69x |
| | PPL | 112.28 | 116.64 | 121.43 | 116.49 | 115.91 |
| PTB-Large | Speedup Rate | 1x | 0.61x | 1.76x | 2.27x | 8.11x |
| | PPL | 78.32 | 80.30 | 82.59 | 80.47 | 80.09 |

Spherical-KMEANS can outperform state-of-the-art methods. This shows that clustering structure of context features is a key to perform fast prediction.

## 7.3 PERPLEXITY RESULTS

Finally, we go beyond top-$k$ prediction and apply our algorithm to speed up the perplexity computation for language models. To get perplexity, we need to compute the probability of each token appeared in the dataset, which may not be within top-$k$ softmax predictions. In order to apply a top-$k$ approximate softmax algorithm for this task, we adopt the low-rank approximation idea proposed in (Shim et al., 2017). For tokens within the candidate set, we compute the logits using exact inner product computation, while for tokens outside the set we approximate the logits by $\tilde{W}h$ where $\tilde{W}$ is a low-rank approximation of the original weight matrix in the softmax layer. The probability can then be computed using these logits. For all the algorithms, we set the rank of $\tilde{W}$ to be 20 for PTB-Small and 200 for PTB-Large. The results are presented in Table 6. We observe that our method outperforms previous fast softmax approximation methods for computing perplexity on both PTB-small and PTB-large language models.

## 7.4 RESULTS ON LARGER VOCABULARY SIZE

In this section we validate that proposed L2S method can still work effectively on models with larger vocabulary sizes. We demonstrate this on Wikitext-103 dataset Merity et al. (2016) by choosing vocabulary size to be 80k. We follow the model and training procedure as in Merity et al. (2016). Precision@1 and Prevision@5 results of L2S are shown in Fig 9. We can see that L2S could speed-up over 15 times without losing much accuracy.

## 7.5 SPEED-UP RATE VERSES VOCABULARY SIZE.

We conduct experiments on PTB-Small setup to study the effect of vocabulary size to speedup. We compress the size of PTB dataset by choosing most frequent words and set the rest of other words to be "unknown" and follow the same training procedure. The cluster size is set to 200 for all setups, and the results are shown in Fig 8. Basically, the results show that the compression rate could be made higher when vocabulary size is larger. This is reasonable as the occurrence of the words follow zipf's distribution in natural languages. So when vocabulary size goes large, we could imagine many words only need to be searched a few times so full softmax computation will not be very efficient. Thus the speed-up rate could be improved further.

## 7.6 QUALITATIVE RESULTS ON CLUSTERING

We selected few clusters and show the words contained in that cluster to demonstrate qualitative results of L2S. Examples are listed in Table 7. Basically, some frequent words will appear in almost all clusters in order to make sure the predicted accuracy is high. For example, "the", "in", and "a"

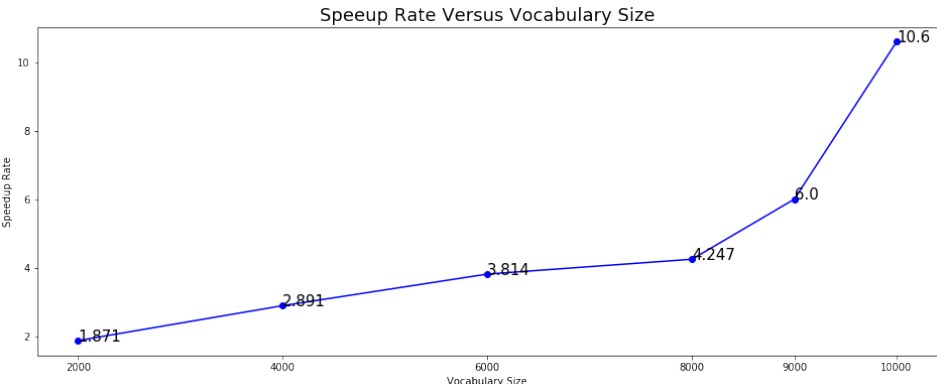

Figure 8:  Speed-up rate versus vobcaulary size of PTB-Small Setup.  Precision@1 for each vocabulary size is selected to be around 0.99.  Precise values are 0.993(2000),0.989(4000),0.988(6000),0.987(8000),0.987(9000),0.998(10000). Values in the parentheses are vocabulary sizes.

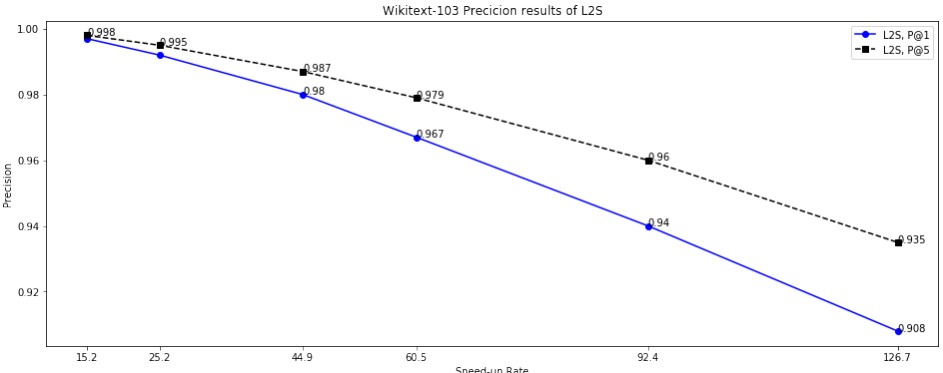

Figure 9:  Precision@1 and Precision@5 versus Speed-up rate of L2S on Wikitext-103 dataset with 80k vocabulary size.

those tokens appear in all clusters shown in the table. On the other hand, each cluster might contain some

In the cluster 1, we can observe that all words related to timing is included. In the second cluster, some terms related to financial investment are included. More interestingly, from the term "over-the-counter" in third cluster we know that it's about finance. And we also observe that important cities of finance world such as Tokyo, "Hong" Kong and London are included. Since these trading trend might be seasonal, we also see that many monthes are included in this cluster.

## 7.7  QUALITATIVE RESULTS ON NEURAL MACHINE TRANSLATION

We select some translated sentences of DE-EN task shown in Table 8 to demonstrate that our algorithm can provide similar translations but with faster inference time.

Table 7: Qualitative results of clustering. We select few clusters to show what might be learned in the clustering.

| Cluster No. | Words |
|---|---|
| 1 | the, \<unk\>, \<eos\>, N, of, to, a, in, and, "s", that, for, is, it, said, on, by, at, as, from, million, with, mr., was, are, he, but, has, will, have, or, they, **year**, which, would, about, says, were, had, up, than, also, if, when, shares, **years**, because, after, could, sales, while, **months**, through, before, analysts, **days**, according, although, compared, **times**, states, april, bankruptcy, per, totaled, **p.m.**, classes |
| 2 | the, \<unk\>, a, in, and, that, for, it, on, at, mr., he, but, they, one, some, we, if, when, i, there, you, **investors**, these, **stocks**, **traders**, **futures**, ual |
| 3 | the, \<unk\>, N, a, that, mr., its, an, new, this, his, one, some, other, may, many, any, **american**, national, recent, her, part, **japan**, early, late, **september**, **august**, **london**, composite, addition, terms, **october**, **july**, **tokyo**, fact, **hong**, response, **january**, **over-the-counter** |

Table 8: Qualitative comparison of our method to full softmax computation. The accelerated model used is the same as reported in Table 3.

| Full-softmax | Our method |
| --- | --- |
| you know , one of the great <unk> at travel and one of the **pleasures** at the <unk> research is to live with the people who remember the old days , who still feel their past in the wind , touch them on the rain of <unk> rocks , taste them in the bitter sheets of plants . | you know, one of the great <unk> at travel and one of the **joy** of the <unk> research is to live **together** with the people who remember the old days , who still feel their past in the wind , touch them on the rain of <unk> rocks , taste them in the bitter sheets of plants. |
| it s the symbol of all **that** we are , and what were capable of as astonishingly <unk> species . | its the symbol of all of **what** we are , and what were capable of as astonishingly <unk> species . |
| when **any of you were** born in this room , there were 6,000 languages **talking** on earth . | when **everybody was born** in this room , there were 6,000 languages **spoken** on earth . |
| a continent is always going to leave out , because the **idea** was that in sub-saharan africa there was no religious faith , and of course there was a <unk> , and <unk> is just the **remains** of these very profound religious thoughts that <unk> in the tragic diaspora of the <unk> . | a continent is always going to leave out , because the **presumption** was that in sub-saharan africa there was no religious faith , and of course there was a <unk> , and <unk> is just the **cheapest** of these very profound religious thoughts that <unk> in the tragic diaspora of <unk> <unk> . |
| so , the fact is that , in the 20th century, in 300 years , it is not going to be remembered for its wars or technological innovation , but rather than an era where we were present , and the massive destruction of biological and cultural diversity on earth either on earth is either **active or** <unk>. so the problem is not the change . | so , the fact is that , in the 20th century , in 300 years , it is not going to be remembered for its wars or technological innovation , but rather than an era where we were present , and the massive destruction of biological and cultural diversity on earth either on earth is either <unk> **or passive**. so the problem is not the change . |
| and in this song , we're going to be able to connect the possibility of what we are : people with full consciousness , who are **aware of the importance** that all people and gardens have to thrive , and there are great moments of optimism . | and in this song , we're going to be able to rediscover the possibility of what we are : people with full consciousness that **the importance of the importance** of being able to thrive is to be able to thrive , and there are great moments of optimism . |

