# OpenReview forum: "Learning to Screen for Fast Softmax  Inference on Large Vocabulary Neural Networks"
_ICLR.cc/2019/Conference_

### Official Review · AnonReviewer2 · 2018-11-01
**I like the pape**

**Rating:** 8
**Confidence:** 4

**Review:**

The paper proposes a way to speed up softmax at test time, especially when top-k words are needed. The idea is clustering inputs so that we need only to pick up words from a learn cluster corresponding to the input. The experimental results show that the model looses a little bit accuracy in return of much faster inference at test time.

* pros:
- the paper is well written.
- the idea is simple but BRILLIANT.
- the used techniques are good (especially to learn word clusters).
- the experimental results  (speed up softmax at test time) are impressive.

* cons:
- the model is not end-to-end because word clusters are not continuous. But it not an important factor.
- it can only speed up softmax at test time. I guess users are more interesting in speeding up at both test and training time.
- it would be better if the authors show some clusters for both input examples and corresponding word clusters.

---

> ### Author Response · Authors · 2018-11-21
> **Response to Reviewer 2**
>
> We are thankful for the constructive comments!!
>
> -- about word clusters are not continuous and training end to end
>
> There are several ways to make word clusters continuous such as using soft clustering, however, these strategies on the other hand will increase the prediction time. Even though word clusters representation is not continuous in L2S, our model can still train end-to-end in the sense that the clustering stage and the label selection are trained jointly with the gumbel technique. Our algorithm back-propagates the gradient to the clustering weights to update both clustering partition and label sets simultaneously.
>
> -- about speeding up training time
>
> We focus on speeding up prediction in this work. We could potentially use the same idea--clustering+learning candidate words, to speed up training as well since we could narrow down the update on a few candidate words instead of the entire vocabulary when updating softmax’s weight matrix. This is certainly an interesting future direction to work on.
>
> -- qualitative examples
>
> We have added two qualitative analyses in the new version. Firstly, we show the words from different clusters learned from our method in Table 7, and observe some interesting structures---some words with similar meanings are in the same cluster. Secondly, examples of translation pairs by our method compared with full softmax results are shown in Table 8.

---

### Official Review · AnonReviewer1 · 2018-11-02
**Fast and accurate approximation to softmax, but more in-depth analysis results would be required**

**Rating:** 6
**Confidence:** 3

**Review:**

This paper presents an approximation to the softmax function to reduce the computational cost at inference time and the proposed approach is evaluated on language modeling and machine translation tasks. The main idea of the proposed approach is to pick a subset of the most probable outputs on which exact softmax is performed to sample top-k targets. The proposed method, namely Learning to Screen (L2S), learns jointly context vector clustering and candidate subsets in an end-to-end fashion, so that it enables to achieve competitive performance.

The authors carried out NMT experiments over the vocabulary size of 25K. It would be interesting if the authors provide a result on speed-up of L2S over full softmax with respect to the vocabulary size. Also, the performance of L2S on larger vocabularies such as 80K or 100K needs to be discussed.

Any quantitative examples regarding the clustering parameters and label sets would be helpful.
L2S is designed to learn to screen a few words, but no example of the screening part is provided in the paper.

---

> ### Author Response · Authors · 2018-11-21
> **Response to Reviewer 1**
>
> We want to thank the reviewer for the useful suggestions!!
>
> -- about larger vocabulary experiment:
>
> We have added an experiment with a much larger dataset --- Wikitext103 with vocabulary size of 80k. The result of prediction time speedup versus accuracy is shown in Figure 9 in the new version. As you can see from the figure, we can achieve more than 15x speedup with accuracy of 99.8%. In addition, in Table 3, we show the result on DE-EN, an NMT task with vocabulary size around 25k. We summarize the vocabulary size of all the datasets in Table 1.
>
> -- about result on speed-up of L2S over full softmax with respect to the vocabulary size
>
> We have included an experiment of prediction time speed-up versus vocabulary size on PTB dataset. Results are summarized in Figure 8. In this figure, we could observe that our method can achieve higher speed-up with larger vocabulary size.
>
> -- about clustering parameters and label sets
>
> We have added Table 7 to show the label sets learned from our method. We observe some interesting clusters---some words with similar meanings are in the same cluster.

---

### Official Review · AnonReviewer3 · 2018-11-06
**a nice method accelerating softmax for prediction in large vocabulary at test time**

**Rating:** 7
**Confidence:** 4

**Review:**

This paper proposes a novel method to speedup softmax computation at test time. Their approach is to partition the large vocabulary set into several discrete clusters, select the cluster first, and then do a small scale exact softmax in the selected cluster. Training is done by utilizing the Gumbel softmax trick.

Pros:
1. The method provides another way that allows the model to learn an adaptive clustering of vocabulary. And the whole model is made differentiable by the Gumbel softmax trick.
2. The experimental results, in terms of precision, is quite strong. The proposed method is significantly better than baseline methods, which is a really exciting thing to see.
3. The paper is written clearly and the method is simple and easily understandable.
Cons:
1. I’d be really expecting to see how the model will perform if it is trained from scratch in NMT tasks. And I have reasons for this. Since the model is proposed for large vocabularies, the vocabulary of PTB (10K) is by no terms large. However, the vocabulary size in NMT could easily reach 30K, which would be a more suitable testbed for showing the advantage of the proposed method.
2. Apart from the nice precision results, the performance margin in terms of perplexity seems not as big as that of precision. And according to earlier discussions in the thread, the author confirmed that they are comparing the precision w.r.t. original softmax, not the true next words. This could raise a possible assumption that the model doesn’t really get the probabilities correct, but somehow only fits on the rank of the words that was predicted by the original softmax. Maybe that is related to the loss? However, I believe sorting this problem out is kind of beyond the scope of this paper.
3. In another scenario, I think adding some qualitative analysis could better present the work. For example, visualize the words that got clustered into the same cluster, etc.

In general, I am satisfied with the content and enjoys reading the paper.

---

> ### Author Response · Authors · 2018-11-21
> **Response to Reviewer 3**
>
> Thanks for your comments and that you enjoyed reading the paper!
>
> Responses to questions:
>
> -- about larger vocabulary experiment:
>
> We have added an experiment with a much larger dataset --- Wikitext103 with vocabulary size to be 80k. The result of prediction time speedup versus accuracy is shown in Figure 9 in the new version. As you can see from the figure, we can achieve more than 15x speedup with accuracy of 99.8%. In addition, in Table 3, we show the result on DE-EN, an NMT task with vocabulary size around 25k. We summarize the vocabulary size of all the datasets in Table 1.
>
> -- about perplexity and probability estimation
>
> This is a great point. We agree that our method tends to generate better approximation of ranking of the words instead of probability of that word. The main reason for the reduced gain for PPL is that to compute PPL, after performing our method (L2S), we need an additional step to assign a probability to words that are not located in the predicted cluster, although this is a rare case (less than 5% chance). There are several potential ways to model this rare case and we chose to use SVD to approximate probability (same as svd softmax [Kyuhong Shim et.al in NIPS 2017]); however, SVD itself has lots of computational overhead. Therefore prediction time speedup is less pronounced for PPL than for the accuracy results.
>
> On the other hand, we get reasonable probability estimation when the word is within the predicted cluster (usually they are top-k predicted words). Therefore we still achieve very good (>10x) speed up in NMT tasks with beam search (see Table 3).
>
>
> -- about qualitative analysis
>
> We have added two qualitative analyses in the new version. Firstly, we show the words from different clusters learned from our method in Table 7, and observe some interesting structures--some words with similar meanings are in the same cluster. Secondly, examples of translation pairs by our method compared with original softmax results are shown in Table 8.

---

### Public Comment · (anonymous) · 2018-11-02
**A few questions**

1) I want to confirm that you used fully pre-trained language/NMT models before learning the softmax approximation.  That is, the context vectors where given and not jointly learned with the approximation?

2) For the perplexity calculation, are you selecting the correct candidate set which contains the ground truth token, and then just using the low-rank approximation for all other words?  Is the probability of a given word reliant on the probability of selecting that candidate set?

3) When defining precision@, you say 'This measures the accuracy of next-word-prediction in LM and NMT'.  However, I don't think that is quite correct.  You seem to be measuring the overlap between the top words matching between the true softmax and the approximation and not if the next word actually matches the ground truth next word?  So even if the true softmax got the word incorrect, you are still trying to match the true softmax.

4) In section 4.2, you say '.5% BLEU'.  I don't think you want the '%' there?

5)  I'm having some difficulty with the notation.    Can you confirm that c_t, c_{ts} and c_{p(h_i), s} are all binary variables?  (also the comma before the subscript s doesn't seem to be used consistently)

Thanks for your time.  I enjoyed this paper.

---

> ### Public Comment · (anonymous) · 2018-11-02
> **Replied to question "A few questions"**
>
> Hi there,
>
> Thanks for your interest and useful clarifying questions !!!
>
> 1) You are right. We didn't train the context vector jointly with approximation. Our problem setup is given a pre-trained NLM, how to speed up the inference operations.
>
> 2) Firstly we need to point out after training the cluster label set (c_t) and clustering weights (v_t), we will just select the cluster by choosing the one with maximal z(h) in eq(2). That is to say, in the inference time, given a hidden state h, the corresponding selected cluster is fixed. Apparently there is no guarantee the ground truth token will be in the selected cluster, but our training objective function tries to make the predicted candidate set contains the ground truth token.
>
>
> 3) Sorry for the confusion, I think we will reconsider how to rephrase the scenario. We are not trying to approximate "next-word-prediction accuracy" but to approximate "next-work-prediction operation".
>
> Since in LM and NMT, next-word-prediction is done by taking the maximal inner product between context vector h and Softmax layer W, we refer "next-word-prediction" as the operation to do so.
> We didn't consider the true "next-word-prediction accuracy" because even for taking the original maximal inner product between softmax W and h, it will only give us around 26% accuracy for P@1 when compared to ground truth token. To increase this accuracy actually means to improve the performance of the model over original W. For this work, we focus on making a given pre-trained LM/NMT faster in prediction time but not making a pre-trained LM/NMT having higher accuracy. Therefore, we try to approximate softmax W (the real operation to generate next word) instead of matching ground-truth label by clustering-based thinking.
>
>
> 4) In section 4.2 and corresponding table 2, we do try to add "%" there. We report the BLEU scores which is within .5% difference when compared to the original BLEU score. For example, in NMT: DE-EN Beam=5 row in table 2 we get 13.4 times speed-up with BLEU score drops from 30.33 to 30.19. If we consider the ratio (30.33 - 30.19) / (30.33) which is around 0.0046 ~= 0.46%. Whereas, ".5" BLEU score would be (0.5)/30.33 ~= 1.65% which is 3 times more loss.
>
>
> 5)  Sorry for the confusion again, we will again consider rephrase the notations. We will check again all notations in particular the comma issue you mentioned. Here, we briefly reply to the dimensions of the notations you mentioned. Let's assume there is |V| vocabularies in the model.
>
> For c_t, it in the shape of |v| x 1 vector and we are trying to make entry either 0 or 1 as a pointer of the inclusion of certain. c_{ts} is s-entry of the c_t vector, and thus is binary in the sense. c_{p_bar{h_i},s} refers to the s-entry of the c_{p_bar{h_i}} vector, p_bar{h_i} defined in the paper is the 1-hot entry of the Straight-Through gumbel, which can be thought as the sampled cluster. Thus  c_{p_bar{h_i}} is a vector of |v| x 1 shape and c_{p_bar{h_i},s} refers to s-entry and yes it's binary eventually.

---

### Author Response · Authors · 2018-11-21
**Summary of Changes**

Hi all,

We appreciate the constructive feedback from the reviewers and the community.  And thanks for the patience for waiting our responses. We have made the following main changes to the current version to make our paper more complete.

1. For NMT task, we apply our method on a new dataset EN-VE translation with vocabulary size of 22749. Results are summarized in Table 3. For this task, our method can achieve 20x speedup with BLEU score of 25.27, and the original softmax’s BLEU is 25.35.

2. Besides additional NMT experiment, we perform our algorithm on a larger vocabulary dataset Wikitext-103, a language model dataset with 80k vocabularies. Results are summarized in Figure 9. For this task, our method can achieve more than 15x speedup with P@1 at 99.8%.

3. We also include an experiment on prediction time speed-up versus vocabulary size on PTB dataset. In this experiment, we vary the vocabulary size and show the speedup and accuracy. Results are summarized in Figure 8, showing that our method achieves higher speed-up with larger vocabulary size.

4. We add two qualitative analysis in the appendix. Firstly, we show the words from different clusters learned from our method in Table 7, and observe some interesting structures--some words with similar meanings are in the same cluster. Secondly, examples of translation pairs by our method compared with full softmax results are shown in Table 8. Please look through those interesting examples!

---

### Meta-Review · Area_Chair1 · 2018-12-13
**Good paper**

**Confidence:** 4
**Recommendation:** Accept (Poster)

**Metareview:**

This paper introduces an approach for improving the scalability of neural network models with large output spaces, where naive soft-max inference scales linearly with the vocabulary size. The proposed approach is based on a clustering step combined with per-cluster, smaller soft-maxes. It retains differentiability with the Gumbel softmax trick. The experimental results are impressive. There are some minor flaws, however there's consensus among the reviewers the paper should be published.